# The Complexity and Significance of Fibroblast Growth Factor (FGF) Signaling for FGF-Targeted Cancer Therapies

**DOI:** 10.3390/cancers17010082

**Published:** 2024-12-30

**Authors:** Anh L. Nguyen, Caroline O. B. Facey, Bruce M. Boman

**Affiliations:** 1Department of Biological Sciences, University of Delaware, Newark, DE 19716, USA; nguyena@udel.edu; 2Center for Translational Cancer Research, Helen F. Graham Cancer Center & Research Institute, 4701 Ogletown-Stanton Road, Newark, DE 19713, USA; 3Department of Pharmacology & Experimental Therapeutics, Thomas Jefferson University, Philadelphia, PA 19107, USA

**Keywords:** fibroblast growth factor, fibroblast growth factor receptor, cancer stem cells, colorectal cancer, breast cancer, lung cancer, cholangiocarcinoma, hepatic cancer, endometrial cancer, bladder cancer

## Abstract

Fibroblast growth factors (FGFs) and their receptors (FGFRs) play a crucial role in cancer development. However, due to their diverse cellular functions, the mechanisms by which they drive cancer are complex. In this review, we discuss the mechanisms of action of FGFs and FGFRs, as well as how the dysregulation of FGF signaling contributes to cancer in various tumor types. The complexity of FGF signaling is partly explained by the large number of FGF isoforms and FGFR receptor types. We provide a classification of FGF ligands based on their signaling modes and binding specificity to FGFRs. Additionally, we explore promising therapeutic strategies being developed to target FGF signaling in oncology, including small molecules, ligand traps, and monoclonal antibody-based inhibitors.

## 1. Background Information

FGFs are a family of multifunctional cell signaling proteins with a wide variety of regulatory, morphological, and endocrine effects on multiple cell types. They function as pluripotent growth factors and play a role in the normal development of animal cells and in tumor metastasis and angiogenesis. The key question is how a wide variety of FGF proteins can bind to and elicit specific responses from different FGFRs. To seek answers, we discuss the FGF and FGFR-based mechanisms of action and how FGF expression becomes dysregulated during cancer development.

One explanation for the pluripotent nature of FGFs is their large number of FGF isoforms and FGFR receptor types. The fibroblast growth factor (FGF) family comprises 22 known isoforms (i.e., ligands) [1], which are classified into seven subfamilies, FGF1, FGF4, FGF7, FGF8, FGF9, FGF15/19, and FGF11 (Table 1) [2], and four transmembrane tyrosine kinase FGF receptors (FGFRs) [1]. Within those 22 isoforms, 18 of them (FGF1–FGF10, FGF16–FGF23) interact with the FGFRs and induce their dimerization [2] to trigger the activation of downstream pathways, such as MAPK, PI3K-AKT, and STAT [3]. The FGF/FGFR signaling is involved and regulates a complex network of cellular proliferation, differentiation, migration, and survival [3].

In recent years, the role of FGF signaling in cancers has caught the attention of researchers and oncologists. The overexpression of FGFs and/or FGFRs in multiple cancers were implicated in tumor progression (Table 2) [4,5]. The oncogenic potential of FGF/FGFR signaling has a tight connection with its physiological ability to promote cell growth, invasion, and apoptosis escapism, which are well-known hallmarks of cancer [5,6]. Moreover, FGF/FGFR signaling is suggested to play a crucial role in cancer metastases by regulating the complex tumor–stromal interactions and the tumor microenvironment to promote angiogenesis [7]. Due to its multifaceted role in various malignancies, FGF/FGFR becomes an attractive target for therapeutic cancer treatments. Several studies were conducted to develop the optimal approach to inhibit this pathway either by small molecule inhibitors, ligand traps, or monoclonal antibodies [8]. Some potent FGFR inhibitors (Section 5) were already approved by the Federal Drug Administration (FDA) for specific cancer types [9]. This article aims to provide a comprehensive overview of the current knowledge about FGF and FGFR structures and physiologies and the molecular mechanisms underlying FGG/FGFR signaling in cancers. Moreover, the latest therapeutic approaches targeting FGF/FGFR signaling will also be discussed.

## 2. The FGF Ligands

The FGF ligands are classified into seven subfamilies based on the coding sequence, protein structure, and biological mode of action, as presented in Table 1 [8,16]. The FGF ligands share between 35% and 60% sequence homology. They typically have 150–300 amino acids in length with a conserved region of 6 identical subunits of 120 amino acid residues [1,16]. Based on the mode of actions, these seven subfamilies of FGF are classified into three groups: intracrine, paracrine, and endocrine (Figure 1).

Five subfamilies of FGFs, FGF1, FGF4, FGF7, FGF8, and FGF9, act as paracrine factors, and are often referred to as canonical FGFs. FGFs are extracellular proteins and interact with the FGFRs on nearby cells (paracrine) or the same cell (autocrine) through binding with heparin or heparan sulfate proteoglycans (HSPGs) [3]. Upon binding with FGFRs, it triggers several downstream pathways, which are involved in cellular proliferation, angiogenesis, and immune responses [12]. Thus, some FGFs become sequestered in the extracellular matrix of tissues, and through this process, may contribute to tumor cell invasion and angiogenesis.

The endocrine subfamily FGF15/19 has evolved distinct mechanisms to function as systemic hormones [24]. These FGFs exhibit a low affinity for heparan sulfate proteoglycans (HSPGs) and activate FGFRs via a different cofactor, Klotho proteins [3,25]. In FGF19, the absence of the conserved glycine-rich motif between the β10 and β12 strands, along with a longer β1–β2 loop in the heparan sulfate-binding region, alters its binding affinity for HSPGs [25]. Similar structural adaptations in other endocrine FGFs, such as FGF21 and FGF23, further reduce their affinity for HSPGs. These modifications allow the FGFs to function as endocrine hormones while still binding FGFRs effectively. The resulting activation triggers signaling cascades similar to those initiated by paracrine FGFs, but at a systemic level, suggesting their critical roles in metabolism, homeostasis, and cancer metastasis.

Intracrine FGFs are intracellular proteins traditionally believed not to interact with FGFRs, and their proposed mechanism of action is through voltage-gated sodium channels, although this idea remains debated [3,26]. These FGFs are also implicated in intracellular signaling via interactions with islet brain-2 (IB2) [27] or the nuclear factor kappa-light-chain-enhancer of activated B cells (NF-κB) essential modulator (NEMO) [28]. Additionally, intracrine FGFs play roles in wound healing and tissue repair, with FGF13 stabilizing microtubules and promoting neuronal polarization, aiding functional recovery after spinal cord injury [29]. In 2023, Biadun and colleagues made a significant discovery, revealing that intracrine FGFs can be secreted into the extracellular matrix through a mechanism similar to that of FGF2, where they bind directly to and activate FGFR1. Under stressful conditions, these FGFs activate FGFR1-dependent pathways, inducing apoptosis in neighboring cells without disrupting their cell cycle [30]. This finding challenges our current understanding of intracrine FGFs, emphasizing the need for further studies to fully elucidate their functional mechanisms.

FGF ligands are also found to express in a tissue-specific manner. In 2010, Tacer and colleagues conducted a study to generate the comprehensive expression atlas in adult mice [26]. While the members in the FGF1 subfamily are found to be highly expressed in the central nervous system (CNS), heart, and kidney, other subfamily’s members, such as FGF4, are found to be highly expressed in the duodenum, colon, and muscle [26,31,32]. Interestingly, the intracrine subfamily is found to have the highest mRNA expression among FGF subfamilies, which suggests they might play an essential role in human physiology [26]. The FGF expression atlas generated by Tacer’s group is mostly consistent with the data obtained from the Human Protein Atlas database, which suggests the complexity of FGF ligand networks through their spatial expression. The difference in sublocation may suggest different physiological significances of each ligand thus potentially play a crucial role in contributing to specific types of cancers. The detail of tissue distribution of FGF ligands obtained from the Human Protein Atlas database is presented in Table 1.

Moreover, FGFs can interact with other growth factors and their receptors, highlighting the complexity of signaling networks. For instance, platelet-derived growth factor receptors (PDGFRs) can form heterocomplexes with FGFR1, and PDGFRβ can directly activate FGFR1 by phosphorylating its tyrosine residues thereby crosslinking signals from both pathways [33,34]. This crosstalk is driven by shared ligands, such as PDGF-BB and FGF2, which can alter ligand activity upon complex formation [33,34]. Additionally, FGFs were shown to interact with Ephrin (Eph) receptors through shared adaptor proteins, like FRS2α, to form complexes that regulate cellular proliferation and differentiation [34]. FGF2 also crosstalks with C-natriuretic peptide (CNP) by inhibiting its effects on cellular proliferation via the MAPK pathway [35], while FGFR3 and CNP exhibit complementary antagonistic actions in ATDC5 cells [36]. These findings underscore the ability of FGFs to regulate cellular responses through interactions with other growth factors, highlighting their biological significance.

## 3. The FGFRs

FGFRs belong to the family of receptor tyrosine kinase (RTK). As of now, there are four known genes that encoded FGFRs: *fgfr1*, *fgfr2*, *fgfr3*, and *fgfr4*. An FGFR contains a highly conserved structure: three extracellular immunoglobulin-like (Ig-like) (D1–D3), a transmembrane domain, and a cytoplasmic kinase domain (Figure 2a) [1,5,37]. Between the two domains, D1 and D2 are the hallmarks of the FGFRs, termed the acidic box, which is an acidic, serine-rich sequence [37]. The ligand binding site lies in the ectodomain in the D2–D3 fragment, while the D1 domain is suggested to play a role in autoinhibition (Figure 2) [38]. Though there are four FGFR genes known, several isoforms of FGFRs exist due to the mRNA alternative splicing in the second half of the D3 domain in FGFR1–FGFR3, which yield **b** (when exon 9 is spliced) (FGFR1b, FGFR2b, and FGFR3b) and **c** (when exon 8 is spliced) isoforms (FGFR1c, FGFR2c, and FGFR3c) [5,37,39] (Figure 2b). These isoforms have distinct binding specificities for FGF ligands as each FGF ligand binds to either the **b** isoform or the **c** isoform, except FGF1 and FGF2, which can bind to both [1,10,37]. The expressions and preferences of these isoforms **b** and **c** are found to be opposite: **IIIb** is mainly expressed in epithelial tissue and prefers mesenchymal tissue-expressed ligands, while **IIIc** is mainly expressed in mesenchymal tissue and prefers ligands expressed in epithelial tissue [40,41]. The ability to alternate the ligand specificity based on **b** or **c** isoforms was suggested to play a role in epithelial—mesenchymal transition (EMT), which is critical in tumor progression as it can lead to the generation of cancer cells with stem cell-like characteristics, or cancer stem cell maintenance [42]. In fact, FGF2, also known as basic FGF (bFGF) due to its higher isoelectric point (positive charge) compared to other FGFs, is recognized as a critical factor in the maintenance of cancer stem cells (CSCs) and the tumor microenvironment due to its ability to bind both FGFR isoforms [43]. In glioblastoma, bFGF is upregulated by JAK/STAT3 activation, and induced by a glioblastoma stem cell-derived protein, Interferon-Induced Transmembrane Protein 3 (IFITM3), which subsequently promotes angiogenesis and metastasis [44]. Additionally, Fessler and colleagues found that bFGF secreted by tumor microvascular endothelial cells can induce CSC marker expression and neurosphere formation in differentiated GBM cells [45]. Moreover, bFGF was shown to enhance epithelial-to-mesenchymal transition (EMT) by inducing cancer-associated fibroblast (CAF) formation, with CAFs being key drivers of EMT due to their predominant overexpression of EMT-related markers such as Vimentin [46]. In breast cancer, bFGF enhances the recruitment of bone mesenchymal stem cells (BMSCs) and their transdifferentiation into CAFs, potentially via the Erk1/2 and Smad3 pathways, leading to increased tumor cell proliferation and collagen production in the tumor microenvironment [47]. Furthermore, bFGF was found to induce the EMT process through YAP1/2 signaling thereby increasing tumor proliferation in gastric cancer [48]. The binding for the partner of each FGFR isoform is summarized in Table 2.

One of the key structures that significantly impacts FGFR function is N-glycans, which cover a substantial portion of the receptor surface [49]. They are primarily located within the extracellular domain, especially the D1, D2, and D3 domains (Table 3) [49]. N-glycosylation influences various aspects of FGFR biology, including FGF-FGFR binding, receptor secretion and stability, and acts as an autoinhibitory mechanism [49]. Specifically, the removal of N-glycans from FGFR1-IIIc increases its binding affinity for FGF2 and heparin-derived oligosaccharides, suggesting that N-glycans negatively regulate the FGF2–FGFR1-IIIc interaction. Additionally, these glycan modifications are crucial for receptor localization and phosphorylation, as demonstrated by the N262Q mutation in FGFR3, which leads to increased localization of oligomannose-type N-glycans in the endoplasmic reticulum and enhanced stability [50]. This mutation also results in upregulated ligands that cause independent phosphorylation of FGFR3 [50]. Furthermore, a study by Gregorczyk and colleagues showed that N-glycans in the D3 domain of FGFR1 do not facilitate its trafficking from the nuclear envelope to the plasma membrane, whereas N-glycans in the D2 domain reduce FGFR1 accumulation in the nuclear envelope [51]. Importantly, the presence of N-glycans in both the D2 and D3 domains is required for FGFR1 trafficking to the plasma membrane [51]. Their findings also revealed that impaired N-glycosylation, leading to FGFR1 accumulation in the nuclear envelope, results in upregulated autoactivation [51]. Together, these results underscore the critical role of N-glycosylation in regulating FGFR kinase activity and its biological and physiological processes.

Among the FGFRs, FGFR4 shares the least homology [52]. FGFR4 has only the **c** isoform, which enhances its specificity towards the ligands [53,54]. In fact, FGFR4 is found to be bound to only either FGF4 or FGF15/19 (Table 2). Interestingly, FGFR4 has a unique amino acid in its kinase domain, specifically in the hinge region, C552, which makes it a more “attractive” target in CRC therapy (Figure 2c) [6]. This unique structure enables a specific FGFR4 inhibitor, which can be promising in cancers that have overexpressed FGFR4, such as colorectal cancer. Some of the FGFR4 inhibitors are developed to covalently bind to the C552 residue of FGFR4, such as BLU-554 (Fisogatinib) [55], FGF401 (Roblitinib) [56], and H3B-6527 [57], that yield promising results in solid tumors.

**Table 3 cancers-17-00082-t003:** Predicted N-glycolysation positions in FGFRs [58,59].

FGFR	Domain	Sequence Position
FGFR1	D1	Asn77; Asn177
	D2	Asn227; Asn240
	D3	Asn264; Asn296; Asn317; Asn330
FGFR2	D1	Asn83; Asn123
	D2	Asn228; Asn241
	D3	Asn265; Asn297; Asn318; Asn331
FGFR3	D1	Asn98
	D2	Asn225
	D3	Asn261; Asn294; Asn315; Asn328
FGFR4	D1	Asn112
	D2	Asn258
	D3	Asn290; Asn311; Asn322

**Figure 2 cancers-17-00082-f002:**
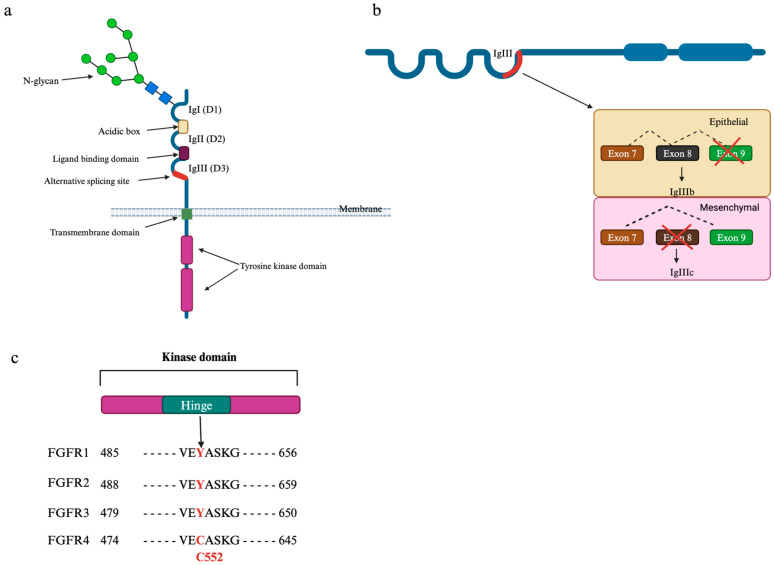
FGFR structures and their alternative splicing sites. (**a**) A single FGFR structure with three Ig-like domains, a transmembrane site, and a tyrosine-kinase domain. (**b**) A schematic to describe the alternative splicing site in the second half of the IgIII domain. Isoform IgIIIb is formed when exon 9 is spliced, and IgIIIc is formed when exon 8 is spliced. IgIIIb is predominant in epithelial cells, while IgIIIc is predominant in mesenchymal cells. (**c**) The hinge region sequence in FGFRs. A different residue (C552) in FGFR4 and the corresponding residue in FGFR1-3 are colored in red. Adapted from Levine et al., 2020 [60].

## 4. FGF/FGFR Signaling

### 4.1. Canonical FGF/FGFR Signaling

FGF ligands in autocrine/paracrine will bind to the FGFRs in a canonical manner through the cofactor heparin or heparan sulfate PHSPG. HSPGs are glycoproteins that contain covalent heparan sulfate (HS) chains, which help shorten the distance of the ligands from the cells [61]. Paracrine FGFs have a relatively high affinity for HSPGs but it is discrete for different ligands [25]. Structural studies found that the HSPG binding site of paracrine FGFs contains the extended β10–β12 region, while the endocrine FGFs (FGF15/19 subfamily) lacks the truncated β10–β12 region, which reduces their affinity for HSPGs [62]. Moreover, crystal studies have revealed the details about the FGF-HSPG-FGFR complex as a 2:2:2 dimer [63]. HSPGs are suggested to stabilize FGF-FGFR binding through engaging them in the dimers thus promoting stable intracellular transduction signals [61,63].

Upon inducing dimerization in FGFR through the help of HSPGs, the receptor subsequentially becomes activated following phosphorylation at FGFR substrate (FRS2a), then FRS2a binds to Src homology 2 (SH2) domain-containing adaptor growth factor receptor-bound protein 2 (GRB2). Subsequently, GRB2 binds to SOS, GRB2-associated-binding protein 1 (GAB1), and the Casitas B-cell lymphoma (CBL) protein through the SH3 domain, triggering a downstream pathway such as Ras, MAPK, and AKT [61,64,65]. The schematic diagram of canonical paracrine/autocrine FGF signaling is depicted in Figure 3a.

Recently, liquid–liquid phase separation (LLPS) was identified as a critical mechanism for FGF signaling, particularly through the bFGF-FGFR axis [66]. Heparin was shown to promote bFGF’s LLPS on the cell surface, resulting in the formation of condensates that enhance FGFR activation [66]. This process is driven by multivalent interactions between bFGF and heparan sulfate proteoglycans (HSPGs), creating a concentrated signaling hub [66]. Once condensed, bFGFs can dimerize and oligomerize, which subsequently influence downstream signaling by recruiting effectors, such as PLCγ1, to the phase-separated compartments [66]. These findings deepen our understanding of canonical FGF-FGFR signaling, highlighting its complexity and the potential for phase separation to serve as a scaffold for enzymatic activities that regulate this pathway.

### 4.2. Non-Canonical FGF/FGFR Signaling

Due to the lack of HSPG binding region in endocrine FGFs, they do not bind to the HSPGs as a cofactor; thus, they can diffuse far away from the cells and act as hormones [67]. Instead, they bind to FGFRs through another cofactor: the Klotho protein [68]. Klotho proteins are single-pass transmembrane proteins, which are homologous to β-glucosidases, and contain KL domains [61]. There are three main Klotho proteins: α-Klotho, β-Klotho, and γ-Klotho. Among these three, α-Klotho and β-Klotho are the cofactors for the endocrine FGFs, particularly facilitating their bindings with the c-splice isoform in FGFR1-3 and FGFR4 [1]. Each endocrine FGF has discrete biological functions: FGF15/19 (FGF15 in mice, FGF19 in the human ortholog) regulates bile acid homeostasis; FGF21 serves as a starvation hormone that initiate stress response; and FGF23 regulates calcium and sodium reabsorption in kidneys [69,70]. Detailed atomic studies have revealed that FGF23 forms a complex with α-Klotho and FGFR1c, while FGF19 activates FGFR4-β-Klotho, and FGF21 forms the FGFR1c-β-Klotho complex [71,72]. The schematic diagram of endocrine FGF signaling is depicted in Figure 3b.

FGFRs can be activated by non-FGF ligands, including lectins, galectins, neural cell adhesion molecule (NCAM), and N-cadherin [2,73]. Specifically, galectins—such as galectin-1, -3, -7, and -8—directly bind to the N-glycan chains on the D3 domain of FGFR1, triggering receptor clustering and FGF-independent activation, leading to downstream effects like apoptosis and cellular division [73]. Unlike canonical FGF ligands, galectins do not require cofactors for FGFR activation, and endogenous galectins can partially inhibit the FGF1-FGFR1 axis [73]. NCAM activates FGFRs through its fibronectin type III domains, which interact with the Ig2–Ig3 region of FGFR1-3, stabilizing FGFR1 by promoting its recycling via Rab11 and Src-dependent mechanisms [2,74]. The clustering of NCAMs during cell–cell adhesion leads to FGFR clustering, which activates the receptor and triggers specific cellular responses [74]. N-cadherin can activate FGFRs in both the presence and absence of FGF ligands, binding to the acid box between the DI and DII domains of FGFRs [75]. In cancer cells, N-cadherin inhibits FGF-dependent internalization of FGFR1, retaining it at the membrane and constitutively activating the Erk1/2 pathway, which promotes tumor invasion [76]. In the absence of FGF ligands, N-cadherin supports neuronal growth through FGFR signaling [77]. Additionally, tumors with high N-cadherin expression show increased Erk1/2 and MAPK activity, which potentiates metastasis [78]. These noncanonical ligands activate FGFRs through distinct mechanisms, illustrating the complexity of FGF signaling in human physiology.

## 5. FGF Signaling in Cancers

Deregulated FGF/FGFR signaling was reported in several types of cancers. The overexpression of FGF signaling is often observed along with poor prognosis and a high rate of treatment resistance. Therefore, FGF signaling as a potential factor in cancer targeted therapies helped develop many approaches. A summary of current approaches targeting FGF signaling is presented in Table 4.

### 5.1. FGF Signaling in Breast Cancer

FGF signaling served as a key player in breast cancer progression, and FGFR1 and FGFR2 are known to be the most common genetic mutations. About 10–15% cases of breast cancers involve FGFR1 amplification, especially in hormone receptor-positive and HER2-negative subtypes [79,80]. FGFR1 amplification is often associated with an increased risk of metastasis and therapy resistance, particularly endocrine therapy [80]. On the other hand, FGFR2 mutations serve as risk factors and increased breast cancer susceptibility [81]. About 3.6% of triple negative breast cancers carry FGFR2 amplification [81] and it is hypothesized to regulate breast cancer development by promoting self-renewal through the NF-kB pathway [81]. FGF alterations in breast cancer are considered valuable prognostic factors. In estrogen receptor-positive breast cancers, FGFR1 amplification correlates with shorter overall survival and a high reoccurrence rate [82]. Moreover, FGFR2 amplification and overexpression are also associated with unfavorable prognosis and treatment resistance [58,83].

FGF-targeting approaches in breast cancer are still in developmental stages. Among these FGFR inhibitors, Erdafitinib, a selective and irreversible pan-FGFR inhibitor, has demonstrated significant antitumor activity in the FGFR2-amplification subset in breast cancer [79]. However, due to the complexity of the FGF/FGFR network, responses to FGFR inhibitors can vary, requiring the identification of groups of patients with specific genotypes to benefit from the therapy. Currently, a combination approach pairing FGFR inhibitors and CDK4/6 inhibitors is being actively studied to improve the therapeutic efficacy in breast cancer [80]. Moreover, the FGF19/FGFR4 axis is suggested to play a critical role in treatment resistance in breast cancer. The overexpression of FGFR4 is observed in breast cancer clones that are resistant to doxorubicin, a common chemotherapy that inhibits DNA relaxation during transcription [84]. Targeting FGFR4 either by inhibiting it with an antagonistic antibody or silencing it with siRNA enhances the efficacy of doxorubicin in FGFR4-expressing breast cancer lines [84]. Moreover, silencing FGFR4 is also found to increase sensitivity to 5-fluorouracil (5-FU) treatment in breast cancers [84]. All these findings highlight the significance of FGF19/FGFR4 in treatment resistance phenomenon in breast cancer; thus, more thorough studies are necessary to explore the clinical significance of the FGF19/FGFR4 axis to improve patient outcomes.

### 5.2. FGF Signaling in Colorectal Cancer

FGF signaling alterations contribute significantly to colorectal cancer (CRC) initiation, progression, and metastasis. FGFR genetic alterations are present in approximately 5–10% of CRC cases, with FGFR1 amplification being the most common [7,85]. The overexpression of FGF ligands, particularly FGF19, was observed in a subset of CRC tumors from the Cancer Genome Atlas (TCGA) database (Figure 4a). We also found that among four FGFRs, only FGFR4 is overexpressed in CRC tumor tissues (Figure 4a), which is consistent with the fact that FGF19 mainly binds to the receptor FGFR4 (Table 2), as depicted in Figure 4b. Upon FGF19 binding, FGFR4 is activated and subsequently activates the downstream pathways, such as AKT and MAPK, to induce cellular invasion [86].

Recently, the FGF19–FGFR4 axis was found to play a crucial role in CRC metastasis [87]. Silencing FGFR4 reduces cellular adhesion in CRC [86], while knocking out FGFR4 in CRC by CRISPR-Cas9 upregulates E-cadherin and downregulates EMT mediators to reduce the tumor metastasis ability [65,86]. Moreover, potential crosstalk between FGFR4 and *Wnt* signaling, a major pathway involved in CRC progression, was elucidated both in vitro and in vivo [88]. Inhibiting FGFR4 by BLU9931 will inhibit the activation of GSK3β/β-catenin in *Forkhead box C1* (*FOXC1*) FOXC1-overexpressed metastatic CRC colonization, which suggests an interaction between FGF and *Wnt* signaling to regulate metastasis, especially when *FOXC1* is strongly associated with metastasis in CRC [88].

FGFR alterations in CRC are often associated with poor prognosis [85]. Patients with high FGF19 expression tend to have a more aggressive disease and shorter overall survival according to our survival analysis from the TCGA database (Figure 4). The prognostic value of FGF signaling in CRC is further complicated by its role in treatment resistance, particularly to anti-EGFR therapies commonly used in metastatic CRC. Treatment approaches targeting FGF in CRC are still in the early stages of development. Several FGFR inhibitors, such as Erdafitinib and AZD4547, are being evaluated in clinical trials for CRC patients with FGFR alterations [9]. Additionally, antibodies targeting FGF19, like fisogatinib, have shown promise in preclinical models and early-phase clinical trials [89]. Combination strategies, pairing FGFR inhibitors with standard chemotherapy or other targeted therapies, are also being explored to enhance efficacy and overcome resistance mechanisms.

### 5.3. FGF Signaling in Cholangiocarcinoma

Cholangiocarcinoma is a rare and aggressive biliary tract cancer. Multiple studies have shown that altered FGF signaling has a significant association with tumor progression in cholangiocarcinoma, especially *FGFR2* fusions with a frequency of approximately 10–15% cases [90]. Besides *FGFR2* fusions, mutations and amplifications of *FGFR1* and *FGF3* and the overexpression of *FGFR4* were also reported in subsets of cholangiocarcinoma cases [91,92]. These *FGFR2* fusions play the most critical role in cholangiocarcinoma as they lead to constitutively active *FGFR2* to subsequently promote tumor growth and survival [65]. Several studies were conducted to identify the partners in these fusions, such as *BICC1*, *TACC3*, *MGEA5*, and *PPHLN* [93,94,95,96]. The presence of FGF signaling alterations is associated with clinical prognosis in cholangiocarcinoma patients. Low-level amplification of *FGFR2* is found to be associated with better overall survival and lower stage tumors [97]. Moreover, the type of *FGFR2* fusion is implied to have different influences on treatment outcomes based on its different protein products [97]. Specifically, there are three subtypes of protein products made from *FGFR2* fusions: (1) classical fusions that retain the tyrosine kinase and the Ig-like domains; (2) sub-classical fusions that only retain the TK domain; and (3) non-classical fusions that lack both [97]. The non-classical fusions appeared to be resistant to FGFR inhibitors, which indicates that the protein products directly influence the therapy results.

Recently, Pemigatinib, a selective FGFR1-3 inhibitor, was approved by the FDA as a treatment for previously treated, unresectable locally advanced or metastatic cholangiocarcinoma with FGFR2 fusions or rearrangements [98]. This approval was based on a Phase 2 study demonstrating an objective response rate of 36% and a median duration of response of 9.1 months [98]. Besides Pemigatinib, there are several FGFR inhibitors, such as Infigratinib and Futibatinib, that also show promising results in clinical trials [9]. These FGFR-targeting therapies propose an encouraging approach to improve outcomes in cholangiocarcinoma, especially in cases where FGFR alterations are detected.

### 5.4. FGF Signaling in Hepatic Cancers

In liver cancer, particularly hepatocellular carcinoma (HCC), FGF19 is found to be notably overexpressed in approximately 5–10% of HCC cases, which implies that FGF signaling plays a crucial role in tumor progression [99]. Similarly to CRC, the overexpression of FGF19 leads to the overactivation of FGFR4 in liver cancer (Figure 4b). Interestingly, in a significant proportion of HCC tumors, FGFR4 is found to be overexpressed even in the absence of FGF19 amplification [74]. Moreover, cases of FGFR1 amplification and FGFR2 fusions are also reported in HCC [100,101]. The overexpression of FGF19 is found to medicate apoptosis resistance in HCC by upregulating the STAT pathway to increase the expression of Il-6 [102]. Recently, another *FGF19*-targeted gene was discovered: *SOX18* [103]. *SOX18* is a profound oncogene in many cancers, and FGF19/FGFR4 was found to upregulate *SOX18* potentially through an interplay with *Wnt* signaling, specifically through the p-FRS2/p-GSK3β/β-catenin axis [101,103]. Tumor growth in *SOX18*-induced HCC metastasis was significantly inhibited after treatment with BLU-9931, a selective FGFR4 inhibitor, which indicates the significance of FGFR4 in metastatic HCC [77,78].

FGF alterations can also serve as a prognostic index in liver cancer. The overexpression of *FGF19* is associated with poor outcomes and a more aggressive disease [99] and *FGFR4* overexpression is also correlated to worse overall survival in HCC patients [99]. Based on its significant involvement in HCC, FGF signaling becomes a promising target in HCC treatments. Fisogatinib (BLU-554), a selective FGFR4 inhibitor, has shown high efficacy in early-phase clinical trials for FGF19-overexpressed HCC patients [104]. Additionally, other FGFR inhibitors, such as Erdafitinib and Lenvatinib, are currently being evaluated in HCC studies. However, due to the complexity of FGF/FGFR signaling and other pathways, combination approaches, pairing FGFR inhibitors with other targeted therapies, are suggested to enhance treatment efficacy and overcome treatment resistance in liver cancer.

### 5.5. FGF Signaling in Lung Cancer

FGF signaling alterations are found to be significantly associated with lung cancer development, particularly in non-small cell lung cancer (NSCLC). Approximately 20% of squamous cell carcinomas and 3–5% of adenocarcinomas have FGFR1 amplification, while about 1–3% of NSCLC cases have FGFR2 and FGFR3 fusions [80]. These genetic alterations lead to overactivated FGF/FGFR signaling, which promotes tumor growth, invasion, and metastasis [80]. For example, the upregulation of FGF signaling is found to be a critical mechanism in which the Janus kinase 2/signal transducer and activator of transcription 3 (JAK2/STAT3) pathway mediates tumor angiogenesis in NSCLC [105,106]. Moreover, multiple studies have indicated that FGF signaling mediates metastasis in NSCCLC [107,108]. A study conducted by Donnem et al. identified that miR-155, an altered angiogenesis-related microRNAs in NSCLC, has a significant correlation with FGF2, which indicates the potential significance of FGF signaling in tumor invasions in NSCLC [107]. The prognostic value of FGF alterations depends on specific cancer subtypes. In squamous cell carcinoma, FGFR1 amplification is found to be associated with poor outcomes [109]. However, FGFR2 and FGFR3 fusions are found to potentially be associated with more favorable outcomes and increased sensitivity to FGFR inhibitors [84].

Currently, many FGFR inhibitors are actively studied and evaluated as a treatment of FGFR alteration subsets in lung cancer, such as erdagitinib, pemigatinib, and infigratinib. Erdafitinib has shown encouraging results for NSCLC patients with FGFR2 and FGFR3 fusions in early-phase clinical trials [9]. However, the efficacy of FGFR inhibitors is not quite promising in FGFR1-amplified NSCLC cases, prompting to perhaps combination strategies [9,110,111]. Thus, more thorough studies are necessary to identify potential combined targeted agents to enhance the efficacy of FGFR-inhibiting approaches in lung cancer.

### 5.6. FGF in Other Cancers

Many other cancer types, including bladder cancer, endometrial cancer, and glioblastoma, are reported to have FGF signaling alterations. In bladder cancer, about 20% of cases have FGFR3 mutations and fusions [112]. In 2019, the FDA approved Erdafitinib as a treatment for locally advanced or metastatic bladder cancer with susceptible FGFR2 or FGFR3 alterations, making a profound advancement in treatment for the FGFR-altered subset of bladder cancer [9].

In endometrial cancer, FGFR2 mutations are the most prevalent, which are found in approximately 10-12% of cases, particularly in endometrioid subtypes [113]. FGFR mutations are implied to be associated with poor outcomes as well as more aggressive tumors in endometrial cancer [114]. Currently, there are some FGFR inhibitors being evaluated in endometrial cancer patients, such as Brivanib, Dovitinib, and Lenvatinib, which show promising results [115].

In brain tumors, FGFR-TACC fusions occur in about 3–7% of glioblastoma cases [116,117]. These fusions lead to constitutive FGFR activation, promoting tumor growth and tumor angiogenesis [90,91]. Although the prognostic significance of FGF signaling in glioblastoma, particularly FGFR-TACC, is still being studied, the prevalence of this fusion in glioblastoma makes them a potential target. A novel FGFR1 inhibitor, CYY292, showed promising results in inhibiting tumor growth by inhibiting the Akt/GSK3β/snail signaling axis in glioblastoma [118]; Pemigatinib was also found to induce a partial response along with neurological improvement [119]. However, extensive studies are necessary to explore the efficacy of FGFR-targeted therapies in glioblastoma, especially in the subset of patients who have FGFR-TACC fusions.

## 6. Other FGF-Targeted Therapies

In addition to small molecule FGFR inhibitors, several therapeutic strategies targeting the FGF/FGFR axis were developed, including FGF ligand traps, monoclonal antibodies (mAbs), antibody–drug conjugates (ADCs), and multivalent conjugates, each offering unique mechanisms of action. FP-1039, a commonly used FGF ligand trap, sequesters FGFs and prevents their binding to FGFR1, inhibiting receptor activation and downstream signaling [120]. This approach avoids the toxicity associated with small molecule inhibitors while effectively reducing tumor growth [120]. Another example is BAY 1179470, a novel mAb targeting FGFR2 isoforms IIIb and IIIc, promoting FGFR2 internalization and degradation thus preventing its activation [121]. mAbs can also be conjugated to cytotoxic agents, as seen in BAY 1187982, a combination of BAY 1179470 and auristatin, which demonstrated potent anti-tumor activity in FGFR2-positive advanced cancers by specifically delivering toxicity to FGFR2-expressing cells [122]. Additionally, multivalent conjugates, such as bispecific ADCs like AfHER2-lFGF2-vcMMAE, target both FGFR1 and HER2 in breast cancer, offering a promising new treatment approach [123]. These bispecific molecules exhibit exceptional cytotoxicity against HER2+ and FGFR+ or dual-positive cancer cell lines, improving specificity and minimizing side effects. Another promising strategy involves synthetic oligonucleotides, such as aptamers, which bind FGFRs to inhibit signaling [124]. These innovative approaches enhance the therapeutic potential of FGF-FGFR signaling in cancer, addressing the limitations of traditional inhibitors and offering improved specificity for more targeted treatments.

## 7. Conclusions

The diverse roles of FGF signaling across different cancer types underscore the importance of comprehensive molecular profiling to identify patients who could benefit from FGFR-targeted therapies. As our understanding of FGF signaling in cancer deepens, novel therapeutic strategies continue to emerge, making FGF a crucial target for new drug development due to its involvement in promoting tumor metastasis and angiogenesis. FGF-targeted therapies hold great promise for advancing cancer treatment, offering a potential factor for the development of more potent and selective FGFR inhibitors, identifying predictive biomarkers for treatment response, and exploring combination therapies to overcome resistance mechanisms and improve patient outcomes across various cancer types.

## Figures and Tables

**Figure 1 cancers-17-00082-f001:**
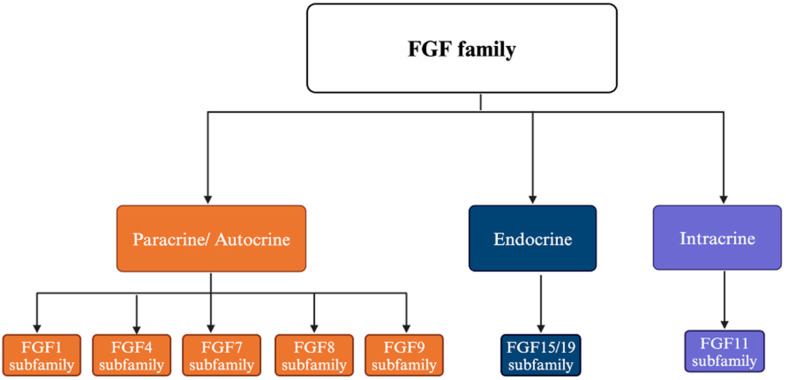
Classification of FGF ligands based on mode of action. (**Orange**): Paracrine/autocrine group; (**Blue**): endocrine group; (**Purple**): intracrine group.

**Figure 3 cancers-17-00082-f003:**
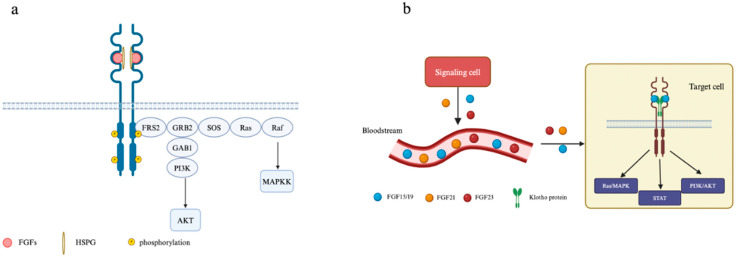
Diagram of paracrine/autocrine and endocrine FGF/FGFR signaling. (**a**) Paracrine/autocrine FGF/FGFR signaling: FGFs bind to HSPGs and the FGFR to induce dimerization of the receptors thus activating them. Activated FGFRs phosphorylate FRS2a, and FRS2 binds to SOS and GAB1 to activate Ras/Raf/MAPK and AKT pathways. (**b**) Endocrine FGF/FGFR signaling: Endocrine FGFs (FGF15/19, FGF21, and FGF23) are secreted and get to the target cell via the bloodstream. A representative of FGF15/19 binds to a Klotho protein and FGFR subsequently triggers downstream pathways, such as Ras/MAPK, STAT, and PI3K/AKT. FGF21 and FGF23 bind to their respective FGFRs in the same manner.

**Figure 4 cancers-17-00082-f004:**
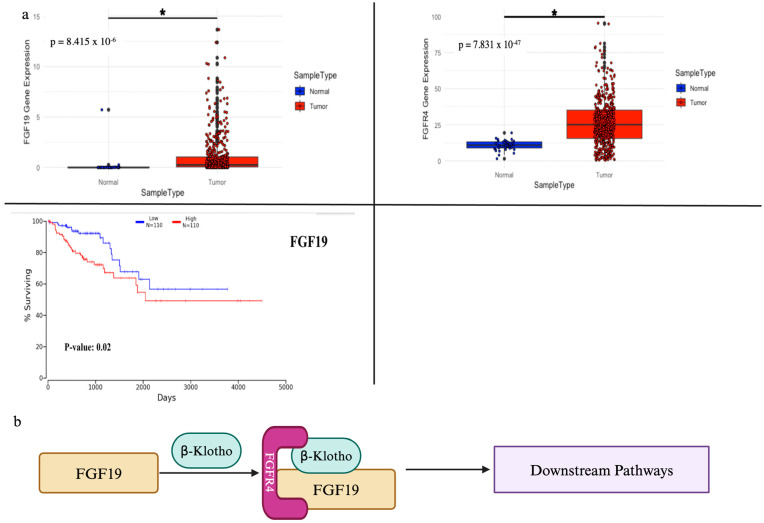
The FGF19/FGFR4 axis in CRC. (**a**) The data were obtained from the TCGA-COAD project. Kaplan–Meier plots summarize the results from the analysis of the correlation between mRNA expression levels and patient survival. Patients were divided based on the level of expression into one of two groups: “low” (under cut off) or “high” (over cut off). The current cut off is at 25% for the lowest (Low) and 25% for the highest (High) expression. *: statistical significance (**b**) A schematic diagram of the FGF19/FGFR4 axis: FGF19 binds to βKlotho and forms a complex with FGFR4. Upon binding, FGFR4 is activated and subsequently activates the downstream pathways.

**Table 1 cancers-17-00082-t001:** Classification and tissue distribution of FGF ligands in humans.

Subfamily	Members	Tissue Distribution
FGF1	FGF1	CNS, kidney, heart, lung
	FGF2	Adipose, heart, lung, aorta, reproductive tissues (both males and females), kidney, liver, colon
FGF4	FGF4	Duodenum, ileum, colon
	FGF5	Cerebellum, gallbladder, colon, small intestine, stomach, kidney
	FGF6	Skeletal muscle
FGF7	FGF3	Cerebellum
	FGF7	Urinary bladder, tongue, skin, reproductive tissues (both males and females), colon, heart, breast
	FGF10	Breast, CNS, reproductive tissues (both males and females), colon, lung, pancreas
	FGF22	CNS, skin
FGF8	FGF8	Skeletal muscles
	FGF17	Brain
	FGF18	Brain stem, spinal cord, lung, heart, skin
FGF9	FGF9	Brain, kidney, retina
	FGF16	Olfactory bulb, heart
	FGF20	Adrenal tissue, thyroid, brain stem
FGF15/19	FGF15/19	Gallbladder, ileum
	FGF21	Liver
	FGF23	Heart, liver, testis
FGF11	FGF11	All tissues
	FGF12	Heart, CNS, spleen
	FGF13	Brain, skeletal muscle, tongue
	FGF14	Brain, cervix

Tissue distribution of FGF ligands data obtained from Human Protein Atlas—Consensus dataset.

**Table 2 cancers-17-00082-t002:** FGF—FGFR binding partners.

Subfamily	FGF Members	FGFRs	Cofactor	Reference
FGF1	FGF1	All FGFRs	Heparin or HSPG	[10]
	FGF2	All FGFRs	Heparin or HSPG	[10]
FGF4	FGF4	FGFR1c, FGFR2c, FGFR3c, FGFR4	Heparin or HSPG	[11]
	FGF5	FGFR3c	Heparin or HSPG	[12]
	FGF6	FGFR1c, FGFR2c, FGFR3c	Heparin or HSPG	[13]
FGF7	FGF3	FGFR3	Heparin or HSPG	[14]
	FGF7	FGFR2b	Heparin or HSPG	[13]
	FGF10	FGFR2	Heparin or HSPG	[15]
	FGF22	FGFR1b, FGFR2b	Heparin or HSPG	[13]
FGF8	FGF8	FGFR2c, FGFR3c	Heparin or HSPG	[16]
	FGF17	FGFR2c, FGFR3c	Heparin or HSPG	[16]
	FGF18	FGFR3	Heparin or HSPG	[17]
FGF9	FGF9	FGFR1b, FGFR2b, FGFR3	Heparin or HSPG	[18]
	FGF16	FGFR1c, FGFR2c, FGFR3c	Heparin or HSPG	[13]
	FGF20	FGFR1	Heparin or HSPG	[19]
FGF15/19	FGF15/19	FGFR4	βKlotho	[20,21]
	FGF21	FGFR1, FGFR2	βKlotho	[22]
	FGF23	FGFR1c, FGFR2c	αKlotho	[22,23]

**Table 4 cancers-17-00082-t004:** FGF-targeted approaches in clinical trials (*data obtained from clinicaltrials.gov*).

Agent	Type of Compound	Target(s)	Cancer(s)	Conditions	Clinical Trial ID	Clinical Status
Anlotinib	Small molecule	VEGFR, PDGFR, pan-FGFR, c-kit	Breast Cancer	Human Epidermal Growth Factor Receptor 2 (HER2)-Negative Breast Cancer Stage II	NCT05558722	Phase 2
AZD4547	Small molecule	Pan-FGFR	Solid Tumors, Lymphomas, or Multiple Myelomas	Refractory tumors and patients must have progressed following at least one line of standard treatment	NCT02465060	Phase 2
E7090	Small molecule	Pan-FGFR	Advanced or Recurrent Solid Tumors	FGFR genetic alterations (including fusion, mutation, amplification)	NCT04962867	Phase 2
Erdafitinib	Small molecule	Pan-FGFR	Urothelial Cancer	Patients with FGFR aberrations who have progressed after 1 or 2 prior treatments, at least 1 of which includes an anti-programmed death ligand 1(PD-[L]1) agent or 1 prior treatment not containing an anti-PD-(L) 1 agent.	NCT03390504	Phase 3
Erdafitinib	Small molecule	Pan-FGFR	Urothelial Cancer	Patients with locally advanced or metastatic urothelial carcinoma, with susceptible FGFR3 whose disease has progressed on or following at least 1 prior line of systemic therapy		FDA approved (2019)
Erdafitinib	Small molecule	Pan-FGFR	Breast Cancer	ER+/HER2−/FGFR-Amplified Metastatic Breast Cancer	NCT03238196	Phase 1
ET0111	Small molecule	Pan-FGFR	Advanced Solid Tumors	Patients with at least 1 lesion, not previously irradiated	NCT05522309	Phase 1
Futibatinib	Small molecule	Pan-FGFR	Advanced Cholangiocarcinoma	Patients with FGFR2 Fusion or Rearrangement		FDA approved (2022)
Futibatinib	Small molecule	Pan-FGFR	Liver Cancer	Advanced or Metastatic FGF19 Positive BCLC Stage A, B, or C Liver Cancer	NCT04828486	Phase 2
HMPL-453	Small molecule	FGFR1, 2, 3	Advanced Solid Tumors	Combined with chemotherapy or anti-PD-1 antibody in patients with advanced solid tumors	NCT05173142	Phase 1b/2
KIN-3248	Small molecule	Pan-FGFR	Advanced solid tumors/Intrahepatic Cholangiocarcinoma/Urothelial Carcinoma	FGFR2 and/or FGFR3 Gene Alterations	NCT05242822	Phase 1
Lenvatinib	Small molecule	VEGFR 1, 2, 3 FGFR1, 2, 3, 4 PDGFR c-Kit	Liver Cancer	Barcelona Clinic Liver Cancer Stage B or C	NCT04368078	Phase 2
LOXO-453	Small molecule	FGFR3	Urinary Bladder Cancer/Other Advanced Solid Tumors	FGFR3 alterations	NCT05614739	Phase 1
Pemigatinib	Small molecule	Pan-FGFR	Cholangiocarcinoma	Previously Treated, Unresectable Locally Advanced or Metastatic Cholangiocarcinoma with FGFR2 Fusion		FDA approved (2020)
Pemigatinib	Small molecule	Pan-FGFR	Bladder Cancer/Non-Muscle Invasive Bladder Cancer	Patients With Recurrent Low- or Intermediate-Risk Tumors	NCT03914794	Phase 2
Pemigatinib	Small molecule	Pan-FGFR	Breast Cancer	HER2-negative Advanced Breast Cancer Patients With FGFR Alterations	NCT05560334	Phase 2
Ponatinib	Small molecule	Pan-tyrosine kinase receptors	Advanced Solid Tumors	FGFR1, FGFR2, FGFR3, FGFR4, RET, KIT mutations	NCT02272998	Phase 2
Ponatinib	Small molecule	Pan-tyrosine kinase receptors	Chronic Myeloid Leukemia (CML)	Chronic phase, accelerated phase, or blast phase chronic myeloid leukemia (CML) Philadelphia chromosome positive acute lymphoblastic leukemia		FDA approved (2024)
Tinengotinib	Small molecule	Aurora A/B, VEGFRs, FGFRs, JAK1/2, and CSF1R	Cholangiocarcinoma	FGFR2 fusion/rearrangement	NCT05948475	Phase 3
TYRA-300	Small molecule	FGFR3	Advanced Urothelial Carcinoma and Other Solid Tumors	FGFR3 gene alteration	NCT05544552	Phase 1/2
SY-4798	Small molecule	FGFR4	Advanced Solid Tumors	Patients must have a histologically or cytologically confirmed advanced solid tumor that is refractory to or unsuitable for standard therapies, or for which no standard treatment options exist.	NCT05498519	Phase 1
ABSK-011	Small molecule	FGFR4	Advanced Liver Cancer	Patients with advanced hepatocellular carcinoma with confirmed overexpression of FGF19	NCT04906434	Phase 1
EVER4010001 (combination with Pembrolizumab)	Small molecule	FGFR4	Advanced Solid Tumors	Patients with confirmed diagnosis of metastatic or locally advanced solid tumors with positive FGF19 in tumor tissues	NCT04699643	Phase 1/2
BB102	Small molecule	FGFR4	Advanced Solid Tumors	Dose escalation part: patients with advanced solid tumors and no available or tolerable standard treatment. Expansion part: patients who have FGF19 or FGFR4 positive advanced HCC or other solid tumors with similar treatment limitations	NCT06258408	Phase 1
Fisogatinib (BLU-554)	Small molecule	FGFR4	Hepatocellular Carcinoma	Confirmed diagnosis of HCC and the patient has unresectable disease and has been previously treated with sorafenib, has declined treatment with sorafenib, or does not have access to sorafenib	NCT02508467	Phase 1
Fisogatinib (BLU-554)	Small molecule	FGFR4	Hepatocellular Carcinoma	Phase Ib: patients have failed after or is unsuitable for the standard systemic therapy against HCC. Phase II: patients have not previously received systemic therapy	NCT04194801	Phase 1b/2
BAY 1179470	Monoclonal antibody	FGFR2	Advanced, Refractory Solid Tumors	FGFR2 positive	NCT01881217	Phase 1
BAY 1187982	Antibody-drug conjugate	FGFR2	Advanced, Refractory Solid Tumors	FGFR2 positive	NCT02368951	Phase 1
FP-1039	FGF ligand trap	FGFR1	Advanced Solid Tumors	Metastatic or locally advanced unresectable solid tumors for which standard curative or supportive measures do not exist or are no longer effective	NCT00687505	Phase 1
FPA144	Monoclonal antibody	FGFR2	Advanced Solid Tumors	Metastatic or locally advanced unresectable solid tumors for which standard curative or supportive measures do not exist or are no longer effective	NCT02318329	Phase 1

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
