# Peer review of "The Complexity and Significance of Fibroblast Growth Factor (FGF) Signaling for FGF-Targeted Cancer Therapies"

_cancers, 2024, doi:10.3390/cancers17010082_

Round 1
Reviewer 1 Report
Comments and Suggestions for Authors
This review comprehensively explores the multifaceted roles of fibroblast growth factors (FGFs) and their receptors (FGFRs). It emphasizes the aberrant expression and mutations of FGFs in cancers, highlighting the distinct roles of various isoforms across different cancer types. Although the authors claim to present the latest advances in various therapeutic strategies, they focus primarily on small molecule-based approaches, with very limited discussion of ligand trap and antibody-based FGF inhibitors.
11. In Table 2, replace "FGF" with "FGFR" in the column for FGFRs binding.
22. There is a typo on line 142 in page 5.
33. The manuscript highlights the FGF19/FGFR4 pathway. To enhance its comprehensiveness, the development of selective FGFR4 inhibitors should be incorporated into Table 3. There are several clinical trials are currently evaluating selective FGFR4 inhibitors, such as BB102 (NCT06258408).
Author Response
WE THANK THE REVIEWERS FOR THEIR HELPFUL COMMENTS.
Reviewer 1
- In Table 2, replace "FGF" with "FGFR" in the column for FGFRs binding.
Answer: Thank you for your comment. We have addressed the typos in the Table 2.
- There is a typo on line 142 in page 5.
Answer: Thank you for your comment. We have addressed the typo on line 142.
- The manuscript highlights the FGF19/FGFR4 pathway. To enhance its comprehensiveness, the development of selective FGFR4 inhibitors should be incorporated into Table 3. There are several clinical trials are currently evaluating selective FGFR4 inhibitors, such as BB102 (NCT06258408).
Answer: Thank you for your comment. We have included trials of selective FGFR4 inhibitors in Table 3.
Reviewer 2 Report
Comments and Suggestions for Authors
The Complexity of Fibroblast Growth Factor Signaling in Cancer: Significance for FGF-targeted Therapies
Comment
1. In this review article, the role of fibroblast growth factor has been well categorized, and a comprehensive explanation is provided. However, in the section on fibroblast growth factor ligands, a more detailed discussion on intracrine mechanisms is required.
2. In lines 116 and 117, the role of b and c isoforms in EMT, tumor progression, and the maintenance of stem cells is explained. However, further clarification on the role and activity of bFGF in cancer stem cells and the tumor microenvironment is necessary to provide more understanding.
3. Does fibroblast growth factor interact with other growth factors too?
Author Response
WE THANK THE REVIEWERS FOR THEIR HELPFUL COMMENTS.
Reviewer 2
- In this review article, the role of fibroblast growth factor has been well categorized, and a comprehensive explanation is provided. However, in the section on fibroblast growth factor ligands, a more detailed discussion on intracrine mechanisms is required.
Answer: Thank you for your comment. We have included the most recent findings on intracrine FGFs in section 2. The FGF ligands.
- In lines 116 and 117, the role of b and c isoforms in EMT, tumor progression, and the maintenance of stem cells is explained. However, further clarification on the role and activity of bFGF in cancer stem cells and the tumor microenvironment is necessary to provide more understanding.
Answer: Thank you for your comment. We have included more information about bFGF and EMT in section 3. The FGFRs
- Does fibroblast growth factor interact with other growth factors too?
Answer: Thank you for your comment. We have included the information about interaction between FGFs and other growth factors in section 2. The FGF ligands
Reviewer 3 Report
Comments and Suggestions for Authors
The manuscript by Nguyen et al focuses on FGF/FGFR and cancer. In general it is well written but lacks several new data that are found in the literature in recent years, thus at this form it is partially outdated and requires improvement in several places:
Page 2, line 75/76: it was demonstrated in few recent studies that so called “intracrine FGFs” can be secreted and, on the cell surface, they directly interact with FGFRs, leding to their activation. This information should be updated in the manuscript.
Pages 4 and 5 and Figure 2. FGFRs are heavily N-glycosylated in their extracellular domains (these N glycans in molecular modeling cover most of FGFR surface) and this modification plays key functions in receptor trafficking, ligand binding and interaction with specific secreted proteins. I would recommend to include this information both in the text and on the figure 2 (showing glycan chains).
Page 5 – the chapter about canonical FGF/FGFR signaling should include recent findings that liquid liquid phase separation (LLPS) by FGF2/FGFR is critical for signaling
Page 6 –the chapter about non-canonical FGF/FGFR signaling should also include information about other non-FGF proteins that can activate FGFRs, especially the ones that can trigger FGFR signaling in fully different way than FGFs do, like galectins
The manuscript should also include at the end, besides FGFR inhibitors, also information about other therapeutic modalities that have been developed to target FGF/FGFRs: FGF ligand traps, mAbs, ADCs, aptamers and other cytotoxic conjugates targeting FGFRs (like multiwalent ones).
Author Response
WE THANK THE REVIEWERS FOR THEIR HELPFUL COMMENTS.
Reviewer 3
The manuscript by Nguyen et al focuses on FGF/FGFR and cancer. In general it is well written but lacks several new data that are found in the literature in recent years, thus at this form it is partially outdated and requires improvement in several places:
Page 2, line 75/76: it was demonstrated in few recent studies that so called “intracrine FGFs” can be secreted and, on the cell surface, they directly interact with FGFRs, leding to their activation. This information should be updated in the manuscript.
Answer: Thank you for your comment. We have included the most recent findings on intracrine FGFs in section 2. The FGF ligands.
Pages 4 and 5 and Figure 2. FGFRs are heavily N-glycosylated in their extracellular domains (these N glycans in molecular modeling cover most of FGFR surface) and this modification plays key functions in receptor trafficking, ligand binding and interaction with specific secreted proteins. I would recommend to include this information both in the text and on the figure 2 (showing glycan chains).
Answer: Thank you for your comment. We have included the most recent findings on N-glycans in section 3. The FGFRs and incorporate the N-glycans in Figure 2.
Page 5 – the chapter about canonical FGF/FGFR signaling should include recent findings that liquid liquid phase separation (LLPS) by FGF2/FGFR is critical for signaling
Page 6 –the chapter about non-canonical FGF/FGFR signaling should also include information about other non-FGF proteins that can activate FGFRs, especially the ones that can trigger FGFR signaling in fully different way than FGFs do, like galectins
Answer: Thank you for your comment. We have included the most recent findings nonconventional FGF ligands in section 4.2. The noncanonical FGF/FGFR signaling.
The manuscript should also include at the end, besides FGFR inhibitors, also information about other therapeutic modalities that have been developed to target FGF/FGFRs: FGF ligand traps, mAbs, ADCs, aptamers and other cytotoxic conjugates targeting FGFRs (like multiwalent ones).
Answer: Thank you for your comment. We have included the other therapeutic modalities in section 6. Discussion and in Table 4.
Round 2
Reviewer 2 Report
Comments and Suggestions for Authors
The revision went well.
Reviewer 3 Report
Comments and Suggestions for Authors
Authors have dealt with issues raised